# Marine Toxins Targeting Kv1 Channels: Pharmacological Tools and Therapeutic Scaffolds

**DOI:** 10.3390/md18030173

**Published:** 2020-03-20

**Authors:** Rocio K. Finol-Urdaneta, Aleksandra Belovanovic, Milica Micic-Vicovac, Gemma K. Kinsella, Jeffrey R. McArthur, Ahmed Al-Sabi

**Affiliations:** 1Illawarra Health and Medical Research Institute, University of Wollongong, Wollongong, NSW 2522, Australia; jeffreym@uow.edu.au; 2Electrophysiology Facility for Cell Phenotyping and Drug Discovery, Wollongong, NSW 2522, Australia; 3College of Engineering and Technology, American University of the Middle East, Kuwait; Aleksandra.B@aum.edu.kw (A.B.); Milica.Micic-vicovac@aum.edu.kw (M.M.-V.); 4School of Food Science and Environmental Health, College of Sciences and Health, Technological University Dublin, D07 ADY7 Dublin, Ireland; Gemma.Kinsella@tudublin.ie

**Keywords:** bioactives, conotoxins 2, Kv1, marine toxins, modulators, potassium channels, sea anemone toxins

## Abstract

Toxins from marine animals provide molecular tools for the study of many ion channels, including mammalian voltage-gated potassium channels of the Kv1 family. Selectivity profiling and molecular investigation of these toxins have contributed to the development of novel drug leads with therapeutic potential for the treatment of ion channel-related diseases or channelopathies. Here, we review specific peptide and small-molecule marine toxins modulating Kv1 channels and thus cover recent findings of bioactives found in the venoms of marine Gastropod (cone snails), Cnidarian (sea anemones), and small compounds from cyanobacteria. Furthermore, we discuss pivotal advancements at exploiting the interaction of κM-conotoxin RIIIJ and heteromeric Kv1.1/1.2 channels as prevalent neuronal Kv complex. RIIIJ’s exquisite Kv1 subtype selectivity underpins a novel and facile functional classification of large-diameter dorsal root ganglion neurons. The vast potential of marine toxins warrants further collaborative efforts and high-throughput approaches aimed at the discovery and profiling of Kv1-targeted bioactives, which will greatly accelerate the development of a thorough molecular toolbox and much-needed therapeutics.

## 1. Introduction

### 1.1. Kv1 Channels

Voltage-gated K^+^ channels (Kv) are intrinsic plasma membrane proteins mediating the selective flow of K^+^ ions down their electrochemical gradient in response to a depolarization in the transmembrane electric field [1]. The selectivity and voltage dependence of Kv channels make them central players in virtually all physiological functions, including the maintenance and modulation of neuronal [2,3,4] and muscular (both cardiac and skeletal) excitability [5,6,7], regulation of calcium signalling cascades (reviewed by Reference [8]), control of cell volume [9,10], immune response [11], hormonal secretion [12], and others.

The Kv channel α-subunit belongs to the six transmembrane (6-TM) family of ion channels (Figure 1a,b) in which the voltage-sensing domain (VSD) formed by transmembrane segments S1–S4 controls pore opening via the S4–S5 intracellular loop that is connected to the pore domain (PD). The PD is formed by transmembrane segments S5–S6 including a re-entrant pore loop bearing the potassium selectivity sequence TVGYG [13]. Depolarization of the transmembrane electric field induces a conformational change in the VSD that leads to channel *activation*, leading to opening of the water-filled permeation pathway permitting K^+^ to flow down their electrochemical gradient. Upon repolarization, the VSD returns to its resting state, closing the channel gate and terminating ionic flow in a process called deactivation. Immediately after deactivation, channels can be reactivated; however, if depolarization-induced channel activation extends beyond a few milliseconds, inactivation ensues, ceasing K^+^ permeability. Kv channels recover from inactivation only after spending enough time at a hyperpolarized potential [14]. The molecular underpinnings of the inactivation processes have been thoroughly examined functionally and structurally, identifying various inactivation types involving distinct and complex molecular mechanisms. Voltage-gated ion channels can inactivate from pre-open closed-states (closed-state inactivation, CSI) or from the open state(s) (open-state inactivation, OSI) [15]. Inactivation can also be categorized depending on the speed of its onset upon activation. In some Kv channels, fast inactivation or N-type inactivation occurs soon after the channel activates and it is mainly due to an intracellular block by the channel’s intracellular N-terminus hence known as the inactivation particle [16]. This process has been directly observed by cryo-electron microscopy (cryo-EM) in a related prokaryotic K channel [17]. In addition to N-type inactivation, a common but relatively slower process happens after tens or hundreds of milliseconds from channel activation that is termed C-type (or slow) inactivation [18]. Even though the extent and complexity of slow inactivation remain the subjects of investigation, the pore structure and the permeating ions appear to play a vital role [19]. Recent structural and functional studies support a mechanism through which the redistribution of structural water molecules accompanies the rearrangement of amino acids within the channel’s inner cavity and outer vestibule, ultimately leading to the collapse of the permeation pathway in C-type inactivation (reviewed in Reference [20]). Modulation of the inactivation process is a powerful strategy to control the cellular availability of Kv channel-mediated currents; thus, both N- and C-type inactivation are responsive to the cellular redox environment [21]. For instance, structural motifs within the Kv channel’s N-terminus/inactivation particle serve as sensors of the cytoplasmic redox potential [22].

K^+^ channels are the most diverse family of ion channels in excitable and nonexcitable tissues, encompassing 40 Kv members allocated into 12 subfamilies: voltage-gated Kv subfamilies, the *Ether-à-go go* (EAG) subfamily, and the Ca^2+^-activated subfamilies [1]. As such, they are implicated in many neurological, cardiac, and autoimmune disorders, which position them as important therapeutic targets [25]. The identified genes for Kv channel α-subunits are classified into twelve subfamilies: Kv1 (Shaker); Kv2 (Shab); Kv3 (Shaw); Kv4 (Shal); Kv7 (KvLQT); Kv10 (HERG); Kv11 (EAG); Kv12 (ELK); and the modulatory “electrically silent” Kv5, Kv6, Kv8, and Kv9 subfamilies (https://doi.org/10.2218/gtopdb/F81/2019.4). The *Shaker*-related Kv1 family is comprised of eight members (Kv1.1–Kv1.8) encoded by the corresponding *KCNA1–KCNA8* genes. Several Kv1 channels have been identified and functionally characterized within their native tissues, exploiting selective blockers (reviewed by References [2,26,27]). The first Kv1 complexes were purified from mammalian brain using the snake venom toxins called dendrotoxins (DTX). These studies indicated that the functional Kv1 channel is a large (Mr ~400 kDa) sialoglycoprotein complex consisting of four pore-forming α-subunits and four cytoplasmically associated auxiliary β-proteins [28] that modulate K^+^ channel activation and inactivation kinetics (for a thorough review, refer to Reference [29]).

The Kv1 channels are expressed in a variety of tissues as homo- or heterotetrameric complexes (Figure 1a,b) [30]. These complexes are formed in the endoplasmic reticulum [31], where monomers are randomly recruited, assembled, and inserted in the plasma membrane [31]. The four cytoplasmic N-terminal domains interact with one another in a strictly subfamily-specific manner, thus providing the molecular basis for the selective formation of heteromultimeric channels in vivo [32,33]. The predominant pathway in tetramer formation involves dimerization of subunit dimers, thereby creating interaction sites different from those involved in the monomer–monomer association during the oligomerization process [34].

In heterologous expression systems, all Potassium Voltage-gated channel subfamily A Member gene (*KCNA*) transcripts encoding Kv1 α-subunits yield functional homo-tetrameric complexes with distinct biophysical and pharmacological profiles [35], (Figure 1c). While theoretically, the combination of different Kv1 subunits could afford impressive functional diversity, only a subset of oligomeric combinations has been elucidated [36,37,38,39,40], suggesting their synthesis and/or assembly are carefully orchestrated. Amongst the K_V_1 channels, Kv1.2 is the most prevalent isoform in neuronal membranes where only a small fraction occurs as a homo-tetramer, while the majority are a hetero-tetramer with other Kv1 α-subunits [36,40]. In these preparations, the less abundant Kv1.1 subunit is consistently identified in oligomers containing Kv1.2 channels.

### 1.2. Mechanisms of Kv Channel Inhibition by Marine Toxins

The diversity of Kv1 channels and their wide distribution, including their specific expression pattern in the central and peripheral nervous systems (CNS and PNS, respectively), together with their vital function in the excitability of nerve and muscle, make them strategic targets of marine toxins. These natural products are synthesized by marine organisms to deter competitors and to aid predation or for self-defense [41]. Many venomous organisms block Kv channel-mediated currents, crippling membrane repolarization, yielding enhanced excitability, and ultimately engendering paralysis in pray or foe [42].

Marine toxins exploit different Kv channel traits to exert their modulatory actions. A commonly used strategy relies on direct occlusion of the narrow potassium permeation pathway from the extracellular side of the channel protein (Figure 2a,b). Toxins inhibiting ionic current via this mechanism are referred to as “pore blockers”. Many structurally and phylogenetically unrelated pore-blocking toxins of Kv channels share a dyad motif composed of a lysine (positive) and a tyrosine/phenylalanine (hydrophobic) [43,44,45]. The lysine residue fits snugly in the Kv channel selectivity filter, sterically occluding K^+^ ion flow, whilst the hydrophobic amino acid in the dyad aids docking and consolidation of the toxin binding. This dyad motif has been proposed to be the minimal core domain of the Kv channel-binding pharmacophore (Figure 2b) [46,47,48,49]. 

Mutational analysis and docking calculations have demonstrated that some marine toxins do not possess the canonical functional dyad or do not seem to use one in the classic “pore blocker” fashion to prevent potassium permeation. In these toxins, a positively charged ring of amino acids participates in electrostatic interactions with the Kv1 outer vestibule. These residues provide surface recognition and anchoring, and concurrently, a network of hydrogen bonds and hydrophobic interactions consolidate “capping” of the channel vestibule, with the peptide toxin acting as a lid over the Kv channel pore (Figure 2b) [50]. 

**Figure 2 marinedrugs-18-00173-f002:**
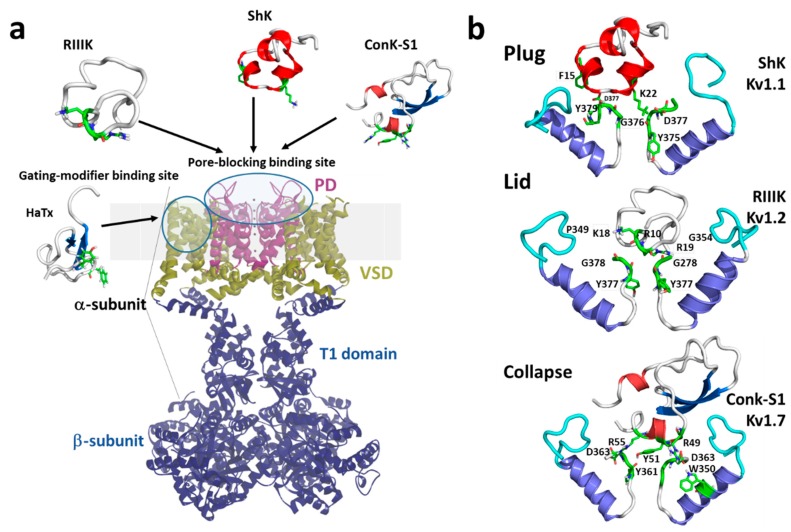
(**a**) Schematic presentation of a side view K_V_1 channel showing the site of interaction with representative pore-blocking peptide toxins from Cone snail (κM-RIIIK, [51] and ConK-S1, PDB: 2CA7, [52]) and sea anemone ShK (PDB: 1ROO, [53]) and gating modifier toxin from spider (HaTx; PDB: 1D1H, [54]). (**b**) The modes of pore blocking (plug, lid, or collapse) illustrated by marine peptide blockers as revealed by the docking models. The outer turret regions (residues 348–359 for Kv1.1, 350–359 for Kv1.2, and 334–343 for Kv1.7) are in cyan, and the inner turret regions (residues 377–386 for Kv1.1, 377–386 for Kv1.2, and 462–469 for Kv1.7) are indicated in green. Only two subunits of the Kv1 channels are shown, for simplicity. Docking was performed using the Haddock webserver [55,56] and the docking model image were generated using Pymol (The PyMOL Molecular Graphics System, [57]).

A distinct mechanism from pore block is achieved by interacting with the gating mechanism of Kv channels. Toxins acting in this fashion are known as “gating modifiers”. The VSD in Kv channels controls pore opening; hence, toxins binding to the extracellularly exposed linker between transmembrane segments S3 and S4, the S3–S4 linker paddle motif within the VSD, inhibit channel function by increasing the energy required to open the channel’s gate by shifting the voltage dependence of activation to more depolarized potentials. Alternatively, some toxins destabilize the Kv channel open state reflected as enhanced entry into a nonconductive inactivated state at potentials where Kv activity would normally be favored [58]. This modulatory mechanism was first shown for Hanatoxin, a gating modifier peptide component of the Chilean rose-hair tarantula venom that inhibits Kv2.1 channels (Figure 2a) [59].

A recently proposed inhibitory mechanism appears as a hybrid strategy between pore blockade and gating modification. A toxin sitting in the channel’s outer vestibule blocks the extracellular side of the permeation pathway and modifies the permeation of water molecules into proteinaceous peripheral cavities in the channel. This creates asymmetries in the distribution of water molecules around the selectivity filter, triggering a local collapse of the channel pore akin to Kv C-type inactivation (Figure 2b) [60].

## 2. Molluscan Peptides that Inhibit Kv1 Channels

Conotoxins constitute a family of small peptide toxins found in the venom glands of cone snails [61]. These marine gastropods of the genus *Conus* are represented by ~800 predatory mollusks [62]. It is believed that the large arsenal of conotoxins within a single venom is used for fast pray immobilization in hunting cone snails [63].

Conotoxins are typically 8–60 amino acid peptides that potently interact with a wide range of voltage- and ligand-gated ion channels and receptors [64]. The cone snail venom peptides evolved to capture their prey (worms, fish, and other mollusks), and their venom is known to interact and modulate several mammalian ion channels with great selectivity [65]. The pharmacological properties of conotoxins have been exploited as molecular tools for the study of mammalian targets [66], and their scaffolds are employed for drug development and potential treatment of human diseases [67]. 

Mature conotoxins are structurally diverse, including disulfide-free and mono- and poly-disulfide-bonded peptides (several reviews deal with the structural diversity of conotoxins; see References [64,68]). Peptides lacking disulfide bonds are flexible, whereas the presence of multiple disulfide linkages provides structural rigidity and provides different three-dimensional conformations depending on the cysteine disulfide framework within the toxin sequence [69]. Cone snail VDPs are often post-translationally modified, including C-terminal amidation, bromination, γ-carboxylation, hydroxylation, O-glycosylation, N-terminal pyroglutamylation, and sulfation [70]. 

Pharmacological classification of the structurally diverse (i.e., cysteine framework/connectivity, loop length, and fold) conotoxins is based on the target type and mechanism of action of the peptides. Twelve pharmacological families are currently recognized (ConoServer [71]). Due to the variable nature of conotoxins, a consensus classification-linking pharmacology to structure has not been agreed upon. Given the nature of this review, we will focus on the pharmacological family classification of the kappa- or κ-conotoxins, which are defined by modulatory activity over potassium-selective channels. The founding member of the κ-conotoxins was identified in the venom of the piscivorous snail *Conus purpurascens* κ-PVIIA by its potent block of *Drosophila* voltage-gated *Shaker* channels [72]. 

Up to now, nine conotoxins are listed as mammalian Kv1 channel blockers in the Kalium database [73]. From those, the activity of Contryphan-Vn from *Conus ventricosus* against Kv1.1 and Kv1.2 was tested by displacement of radiolabeled *Bunodosoma granulifera* Kv1 blocker (BgK), showing weak activity at 600 µM [74]. Therefore, Contryphan-Vn modulatory activity against Kv1 channels remains to be verified.

The other κ-conotoxins listed belong to various structural families of disulfide-rich peptides (A, I, J, M, O, and the Conkunitzins; Figure 3 and Table 1). Disulfide-rich κ-conotoxins have been shown to act as pore blockers using canonical interactions through the “functional dyad” and the “ring of basic residues” as molecular determinants of κ-conotoxin modulation of Kv1 channel conductance. Such mechanisms of action have been described in scorpion and cnidarian VDP toxins blocking Kv1 channels; hence, κ-conotoxins share important features that enable Kv1 channel inhibition in a similar way to other animal VDP blockers.

Despite the great abundance of *Conus* peptides characterized to date, relatively few have been shown to interact with Kv channels. κM-RIIIK from *Conus radiatus* [77] (Figure 2 and Figure 3) is 24 residues long, and it is structurally homologous to the well-known voltage-gated sodium channel blocker µ-GIIIA [78]. RIIIK was originally identified as a Shaker (*Drosophila*) and TSha1 (trout) Kv1 orthologue channel blocker [79]. Later, RIIIK became the first conotoxin described to modulate human Kv1 channels, selectively blocking homomeric Kv1.2 without apparent effects over Navs or mammalian homologs Kv1.1, Kv1.3, Kv1.4, Kv1.5, and Kv1.6 recorded by Two Electrode Voltage Clamp recording (TEVC) in *Xenopus* oocytes [77]. Interestingly, heteromerization with Kv1.2 α-subunits suffices to render Kv1.1, Kv1.5, and Kv1.7, containing heterodimeric channels sensitive to low micromolar RIIIK [80]. 

Binding of κM-RIIIK to closed (deactivated) Kv1.2 channels is ~2-fold stronger than to the open state, hinting towards state-dependent interactions between this peptide and the Kv1s. Importantly, RIIIK blocks its Kv1 channel targets, through a pharmacophore comprised of a ring of positive charges and not via the classical “dyad motif” [51,81].

### 2.1. κM-RIIIJ

Further analyses of the venom of *Conus radiatus* revealed a second, closely related peptide, named κM-RIIIJ, that displayed 10-fold higher potency (~30 nM) blocking homomeric Kv1.2-mediated currents. Comparison of RIIIK and RIIIJ activity in an animal model of ischemia/reperfusion revealed that the latter was cardioprotective, an effect adjudicated to RIIIJ’s higher potency at inhibiting heterodimeric Kv1-mediated currents [80]. 

An in-depth evaluation of RIIIJ was performed against heteromeric channels generated by a covalent linkage composed of Kv1.2 and all other Kv1s subunits (except for Kv1.8) at different stoichiometries and arrangements [82]. This work revealed that RIIIJ exquisitely targets asymmetric Kv channels composed of three Kv1.2 subunits and one Kv1.1 or Kv1.6 subunit. RIIIJ’s apparent affinity for the asymmetric complex is ~100-fold higher than for the homomeric Kv1.2 complex. 

Recently, the discerning sensitivity of RIIIJ to its heteromeric Kv1 channel target was exploited to comprehensively classify and characterize individual somatosensory neuronal subclasses within heterogenous populations of dorsal root ganglion (DRG) neurons [83]. RIIIJ’s selectivity was used to distinguish two functional Kv1 complexes in mouse dorsal root ganglion (DRG) neurons. One being RIIIJ’s high-affinity target (3 × Kv1.2 + Kv1.1 or Kv1.6), and the second component characterized by inhibition at higher RIIIJ concentrations arguably composed of homo-tetrameric Kv1.2 subunits [82]. The functional behavior of large DRG (L-DRG) neurons exposed to RIIIJ was used to classify L-DRGs in six discrete neuronal subpopulations (L1–L6). Interestingly, this peptide’s block of heteromeric Kv1 channels in subclass L3 and L5 neurons lead to enhanced calcium signals consistent with their contribution to repolarization after a depolarizing stimulus, whilst in subclass L1 and L2 neurons, exposure to RIIIJ decreased the threshold for action potential firing. The integration of constellation pharmacology [66], electrophysiology, and transcriptomic profiling using RIIIJ as a pharmacological tool served to functionally assess three biological levels spanning the molecular target Kv1 channel (Kv1.2/Kv1.1 heteromer), the functional characteristics of specific neuronal subclasses and the physiological system (i.e., proprioception) in which they participate.

**Table 1 marinedrugs-18-00173-t001:** Some characteristics of known conotoxins targeting the Kv1 channel.

Conopeptide	Source	Family	Target Channel(s) (IC_50_)	References
CPY-Pl1	*C. planorbis*	CPY	Kv1.2 (2 μM); Kv1.6 (170 nM)	[84]
CPY-Fe1	*C. ferruginesus*	CPY	Kv1.2 (30 μM); Kv1.6 (8.8 μM)	[84]
κM-RIIIJ	*C. radiatus*	M	hKv1.2 (33 nM)	[80]
κM-RIIIK	*C. radiatus*	M	hKv1.2 (300 nM) rKv1.2 (335 nM)	[79]
Pl14a (κJ-PlXIVA)	*C. planorbis*	J	hKv1.6 (1.6 μM)	[76]
κ-ViTx	*C. vigro*	I2	rKv1.1 (1.6 μM) rKv1.3	[85]
Conkunitzin-S1	*C. Striatus*	Conkunitzins	Kv1.7 (< nM)	[12]

### 2.2. Conk-S1

Conk-S1 (Conkunitzin-S1; Figure 2 and Figure 3) was the first reported member of a novel family of marine toxins characterized by the Kunitz structural fold [52]. Conk-S1 is a 60-residue marine toxin from the venom of *Conus striatus* that blocks Shaker [52] and mammalian Kv1 channels [12]. The crystal structure of Conk-S1 displays a Kunitz-type fold in which an NH_2_- terminal 3-10 helix, 2-stranded β-sheet, and the COOH- terminal α-helix are stabilized by 2 disulfide bridges and a network of non-covalent interactions [86]. Despite having only two of the three highly conserved cysteine bridges present in canonical Kunitz peptides, such as bovine pancreatic trypsin inhibitor (BPTI) and Dendrotoxin-κ, structural analyses using NMR spectroscopy verified the presence of the Kunitz-type fold not only in Conk-S1 but also in its homolog Conk-S2 [87]. 

A comprehensive selectivity profile amongst Kv1, Kv2, Kv3, Kv4, BK and EAG channels is available for Conk-S1 [12]. With such information in hand, it was possible to utilize Conk-S1 as a pharmacological tool to identify the role of Kv1.7 channels in glucose-stimulated insulin secretion (GSIS) in pancreatic β cells [12]. Conk-S1 not only was useful as a molecular tool but also was shown to enhance insulin secretion ex vivo in a glucose-dependent manner in islets of Langerhans as well as in vivo in anesthetized rats. Rats treated with Conk-S1 did not evidence any adverse effects, highlighting the potential of Conk-S1 as a therapeutic scaffold for the treatment of hyperglycemia related disorders. 

Mutation of the *Shaker* K^+^ channel residue K427 to aspartate enhances Conk-S1 potency of block >2000-fold, suggesting Conk-S1 interactions with the Kv channel vestibule (see Figure 2b) [52]. Recent structural, functional, and computational work proposes a novel mechanism of the block for K^+^ channel blockers [60]. Conk-S1 does not seem to directly block the ion conduction pathway, but instead, its binding causes disruptions in the structural water network responsible for the stabilization of the Kv channel activated state [88], causing the collapse of the permeation and the consequent hindrance of K^+^ ion flow, similar to what has been described to occur during slow inactivation [60]. This systematic and elegant work was performed on Kv1.2 channels (Conk-S1, IC_50_: 3.4 ± 1.3 µM) instead of the previously described highest affinity mammalian target Kv1.7. The affinity of Conk-S1 for human Kv1.7 channels is 37 ± 5 nM (*Xenopus* oocytes [89]) and 439 ± 82 nM for the mouse orthologue determined in mammalian cells [12]. Interestingly, comparison of the pore sequences of Kv1.1–Kv1.6 and Shaker channels lead to the conclusion that Conk-S1’s preferential toxin action against the Drosophila channel (502 ± 140 nM, *Xenopus* oocytes) was dominated by aromatic interactions mediated by a phenylalanine in Shaker position 425, whilst in hKv1.7, a histidine (341) is present in the equivalent position. 

The observation that heteromerization with Kv1.7 enhances Conk-S1 affinity towards Kv1.2 containing hetero-multimeric Kv channels [12] suggests that such a mechanism of block would extend to Kv1.7-mediated current inhibition as well as to other homo and hetero-tetrameric Kv1 channels.

### 2.3. κ-PVIIA

The venom of *Conus purpurascens* is a source of the founding member of the kappa conotoxins κ-PVIIA (or CGX-1051). PVIIA is a 27-amino-acid-long peptide that potently blocks Shaker potassium channels [72,75]. PVIIA was reported to reduce myocardial lesions in rabbit, rats, and dogs, exhibiting protective effects relevant to ischemia/reperfusion-induced cardiomyocyte damage [90]. In these animal models, acute intravenous administration of PVIIA substantially reduces myocardium infarct size without adverse alterations in cardiovascular hemodynamics [91]. However, attempts to identify a mammalian target of PVIIA have been unsuccessful with 2 µM PVIIA failing to inhibit Kv1.1 or Kv1.4-mediated currents expressed in *Xenopus laevis* oocytes and recorded by two-electrode voltage clamp [92].

While the mechanism underlying the cardioprotective efficacy of κ-PVIIA is unclear, the reported preclinical results in animal models of ischemia/reperfusion suggest that κ-PVIIA may represent a valuable adjunct therapy in the management of acute myocardial infarction [93].

### 2.4. κ-ViTx 

The structural superfamily I2 of *Conus* peptides was established with the discovery of κ-conotoxin ViTx in the venom of *Conus virgo*. In ViTx, four disulfide bridges crosslink a chain of 35 amino acids. TEVC recordings in *Xenopus* oocytes showed that ViTx inhibits voltage-gated K^+^ channels rKv1.1 (IC_50_: 1.59±0.14 μM) and hKv1.3 (IC_50_: 2.09±0.11 μM) but not Kv1.2 (up to 4 μM). Activity on other Kv1 channels has not been reported [85].

### 2.5. SrXIa

SrXIa was purified from the venom of vermivorous *Conus spurius* and was found to inhibit Kv1.2 and Kv1.6 without apparent effects over Kv1.3 channels. SrXIa does not contain lysine residues and thus is considered to lack a functional dyad to support Kv1 channel blockade. Moreover, a ring of arginine including R17 and R29 were shown to be important for its biological activity [94]. Activity on other Kv1 channels is missing.

### 2.6. Promiscuous Conotoxins Interacting with Kv1 Channels

#### 2.6.1. pl14a

The J-conotoxin pI14a isolated from the vermivorous cone snail *Conus planorbis* is 25 amino acids long, from which six residues form an elongated NH_2_- terminus and four cysteines are bonded into the “1-3, 2-4” connectivity, and is decorated with a C-terminal amide group (Figure 3). NMR structure determination revealed one α-helix and two 3(^10^)-helices stabilized by two disulfide bridges [76]. pl14a inhibits Kv1.6 channels (IC_50_ = 1.59 µM) as well as neuromuscular α1β1εδ and neuronal α3β4 nicotinic acetylcholine receptors (IC_50_s = 0.54 µM and 8.7 µM, respectively). Importantly, 1-5 μM pl14a had negligible effects over Kv1.1−Kv1.5, Kv2.1, Kv3.4, Nav1.2, or N-type presynaptic Cav channels [76]. An interesting feature of pl14a is that it contains a putative Kv channel blocking “dyad” formed by residues K18 and T19 as well as a ring of basic residues consisting of R3, R5, R12, and R25. In silico predictions suggest that pI14a inhibition of Kv1.6-mediated currents is mainly supported by the basic ring of amino acids [95]; however, this awaits experimental verification. 

#### 2.6.2. Tyrosine-Rich Conopeptides CPY-Pl1 and CPY-Fe1

The conopeptide family Y (CPY) was defined by the discovery of two 30-amino-acid-long peptides, named CPY-Pl1 and CPY-Fe1, found in the venoms of vermivorous marine snails *Conus planorbis* and *Conus ferrugineus*. VDPs belonging to this family do not contain disulfide bridges and appear unstructured in solution. Nevertheless, the NMR analysis of CPY-PI1 revealed a helical region around residues 12–18 [84]. 

Functionally, both peptides are more active against Kv1.6 and Kv1.2 than Kv1.1, Kv1.3, Kv1.4, and Kv1.5, with CPY-PI1 displaying ~50-fold more potency against Kv1.6 than CPY-Fe1 (IC_50_ 0.17 µM and 8.8 µM, respectively), being ∼18-fold more potent for pl14a. At 1 μM, these peptides also inhibit currents mediated by N-methyl-D-Aspartate (NMDA) receptors (NR1–3b/NR2A and NR1–3b/NR2B) and Nav1.2 channels. Anecdotally, devitellinized oocytes exposed to hydrophobic CPY peptides become “leaky”, suggesting that these peptides could intercalate into the plasma membrane either to destabilize it or to perhaps display pore-forming activity [84]. 

#### 2.6.3. µ-PIIIA

Conotoxin μ-PIIIA and μ-SIIIA inhibit mammalian Nav1.2 and Nav1.7 channels with nanomolar potency [96] and bacterial sodium channels NaChBac and NavSp1 in the picomolar range [97]. It has been recently shown that these μ-conopeptides can also selectively inhibit Kv1.1 and Kv1.6 channels with nanomolar affinity while sparing other Kv1 and Kv2 family members [96]. Functional evaluation of chimeras between μ-PIIIA sensitive and insensitive isoforms revealed that these toxins interact with the Kv pore region with subtype specificity largely determined by the extracellular loop connecting the channel pore and transmembrane helix S5 (turret). 

In contrast to all other pore-blocking κ-conotoxins, the binding of μ-PIIIA to Kv1.6 channels reaches equilibrium after several tens of minutes, pointing towards an alternative Kv1 mechanism of inhibition for μ-PIIIA. Docking and molecular dynamics simulations were used to assess the interaction between µ-conotoxins and Kv1 channels [98]. This work proposed similar binding modes of µ-PIIIA to Kv1.6 and Kv1.1 homomeric channels supported by hydrogen bonding between R and K residues from µ-PIIIA’s α-helical core and the central pore residues of the Kv channel. In such circumstances, effective pore blockage would occur by dual interaction of the μ-conotoxin with both inner and outer pore loops of the Kv channel. This implies that the composition of the channel inner pore loop determines the orientation of the µ-PIIIA, which is further consolidated by hydrogen bonding with the Kv1 extracellular pore loops. The subtype specificity of µ-PIIIA among the Kv1 family members was then rationalized by unfavorable electrostatic interactions between charged residues in the pore loops of the µ-PIIIA-resistant Kv1s.

The apparent binding kinetics of μ-PIIIA to Kv1.6 channels were too slow to allow estimation of potency from concertation response curves as it is customary for potassium channels blocking peptides. This poses the question of whether common peptide screenings on Kv channels performed by relatively short (~5 min) exposures to the toxins are consistently missing positive hits. Alternatively, binding equilibrium determinations in experiments that extend over tens of minutes may be providing an overestimation of potency due to intrinsic confounding factors (such as current rundown and cell viability) inherent to the biological system and experimental conditions used.

#### 2.6.4. κP-Crassipeptides

From Crassispirine snails, a group of venomous marine gastropods, κP-crassipeptides were isolated [99]. Three peptides were characterized to be Kv1 channel blockers, CceIXa, CceIXb, and IqiIXa. The same study showed that, among the tested neuronal hKv1 channels, CceIXb was selective for Kv1.1 with IC_50_ ~ 3 μM. In 1 mM concentration, the other two toxins did not elicit any detectable effects when tested on these Kv1 targets [99]. However, CceIXa and b peptides elicited an excitatory phenotype in a subset of small-diameter capsaicin-sensitive mouse DRG neurons that were affected by the Kv1.6 blocker κJ conotoxin pl14a [94,99]. Since κJ conotoxin pl14a is broader in selectivity among Kv1 channels expressed in DRGs, CceIXa might be more selective for particular combinations of heteromeric Kv1 channels. 

## 3. Cnidarian Peptides that Inhibit Kv1 Channels

Sea anemones (phylum Cnidarian) produce various classes of peptide toxins targeting a diverse array of ion channels that serve the functions of defense from predators and immobilization of potential prey [100]. Some marine toxins found in sea anemones target Kv1 channels in which block leads to neuronal hyperexcitability and muscle spasms. These marine toxins have been shown to have important therapeutic applications in the treatment of autoimmune diseases including multiple sclerosis, rheumatoid arthritis, and diabetes [101,102]. 

Due to the large diversity of toxins produced from sea anemones and both their functional convergence and promiscuity, classification of sea anemone toxins has proven difficult. A recent review has attempted to circumvent this by classifying sea anemone proteinaceous toxins into three major groups: (1) enzymes, (2) nonenzymatic cytotoxins, or (3) nonenzymatic peptide neurotoxins [103]. The Kv channel targeting sea anemone toxins all fall into the third group, peptide neurotoxins, which can be further classified into 9 structural families. To date, of these subfamilies, only six have a Kv-selective toxin representative (ShK, Kv type 1; Kunitz-Domain, Kv type 2; B-Defensin-like, Kv type 3; Boundless β-hairpin (BBH), Kv type 4; Inhibitor Cystine-Knot (ICK), Kv type 5; and Proline-hinged asymmetric β-hairpin (PHAB), Kv type 6; see Table 2).

The anemone VDP toxins interact with Kv1 channels, are typically 17–66 amino acids long, and are cross-linked by 2–4 disulfide bridges [104,105]. Up to 21 sea anemone toxins are listed as mammalian Kv1 channel blockers in the kalium database, which populate all Kv types, except Kv type 5. Each of these types of cnidarian toxins are examined in detail below.

**Table 2 marinedrugs-18-00173-t002:** Sea anemone peptides directed against Kv1 channel.

Toxin	Source	Inhibited Kv1 Channels	References
**Type 1**			
ShK	*Stichodactyla helianthus*	Kv1.1, Kv1.3, Kv1.4, 1.6	[106,107]
AeK	*Actinia equina*	^125^I α-DTX binding to synaptosomal membranes (IC_50_ 22 nM)	[108]
AETX K	*Anemonia erythraea*	^125^I α-dendrotoxinDTX binding to synaptosomal membranes (IC_50_ 91 nM)	[109]
AsKS	*Anemonia sulcata*	Kv1.2	[110,111]
BcsTX1/2	*Bunodosoma caissarum*	BcsTx1 Kv1.2, Kv1.6 BcsTx2 Kv1.1, Kv1.2, Kv 1.3, Kv1.6, Shaker IR with nM IC_50_	
BgK	*Bunodosoma granulifera*	Kv1.1, Kv1.2, Kv1.3, Kv1.6	[112,113]
HmK	*Heteractis (Radianthus)* *magnifica*	Kv1.2, Kv1.3	[114,115]
**Type 2**			
AsKC1	*Anemonia sulcata*	Kv1.2	[111]
AsKC2	*Anemonia sulcata*	Kv1.2	[116]
AsKC3	*Anemonia sulcata*	Kv1.2	[116]
APEXTx1	*Anthopleura elegantissima*	Kv1.1	
SHTXIII	*Stichodactyla haddoni*	^125^I α-DTXdendrotoxin binding to synaptosomal membranes (IC_50_ 270 nM)	[117]
**Type 3**			
BDS-I	*Anemonia sulcata*	Kv1.1–5 < 20% inhibition at 10 µM	[116]
APETx1/2/4	*Anthopleura elegantissima*	Kv1.1-6 < 30% inhibition at 100 nM	
PhcrTx2	*Phymanthus crucifer*	Slight inhibition on DRG Kv currents at µM concentrations	[118,119]
**Type 4**			
SHTX I/II	*Stichodactyla haddoni*	None	
**Type 5**			
BcsTx3	*Bunodosoma caissarum*	Kv1.1, Kv1.2, Kv 1.3, Kv1.6, Shaker IR	[110]
PhcrTx1	*Phymanthus crucifer*	Slight inhibition on DRG Kv currents at µM concentrations	[120]
**Type 6**			
AbeTx1	*Actinia bermudensis*	Kv1.1, Kv1.2, Kv1.6, Shaker IR	[121]

### 3.1. Kv Type 1 Anemone Toxins

Kv type 1 toxins are toxins that include an ShK motif identified from stichodactylatoxin ShK extracted from *Stichodactyla helianthus*. Other VDPs that fall in this family include AeK (*Actinia equina*), AETX K (*Anemonia erythraea*), Kaliseptine AsKS (*Anemonia sulcata*), BcsTXI/II (*Bunodosoma caissarum*), BgK (*Bunodosoma granulifera*), and HmK (*Heteractis magnifica*). They are composed of 34–38 amino acids and cross-linked by three disulfide bridges (3–35, 12–28, and 17–32) [100]; see Figure 4.

#### 3.1.1. ShK

One of the first Kv1 channel blockers characterized was ShK (*Stichodactyla helianthus* K^+^ channel toxin; Figure 2 and Figure 4a [124,125]. ShK potently blocks Kv1.3 and Kv1.1 over Kv1.4 and Kv1.6 channels [106,107]. The amount of ShK found in the *Stichodactyla helianthus* body is relatively small, yet chemical synthesis of the wild-type peptide and its analogs allowed its in-depth study. ShK is a 35-amino-acid peptide with a molecular mass of 4055 Da containing three disulfide-bonded cysteine pairs (C3–C35, C12–C28, and C17–C32) [53,125]. Surface residues of ShK bind at the entrance of the Kv1 channel and block ion conduction by plugging the pore using Lys22 (Figure 2b). The position of the two key binding residues (K22 and Y23) in ShK is conserved in related K^+^ channel blocking peptides from other sea anemones (Figure 4b) [46]. Alanine scanning experiments also identified three other amino acids, S20, K22, and Y23, as essential for the binding of ShK to rat brain potassium channels [107]. In T lymphocytes, Kv1.3 channel activity seems to dominate the membrane potential, where high potency block of this current by ShK highlights its potential use as an immunosuppressant [106,107,126]. However, this peptide has strong binding affinity for neuronal Kv1.1 as well as for its bona fide target Kv1.3 in effector-memory T cells [127]. Thus, the identification of ShK analogs that are highly selective for Kv1.3 over Kv1.1 would enable their use in the treatment of autoimmune diseases such as rheumatoid arthritis and diabetes [102]. To address this, much effort has been dedicated to the optimization of ShK’s sequence to bias selectivity towards Kv1.3. For example, the N-terminal extension on ShK (EWSS) is 158-fold more selective to Kv1.3 over Kv1.1 [127]. Non-peptide-based modifications of ShK include the addition of a 20 kDa poly(ethylene glycol) in ShK-PEG, which increased ShK selectivity 1000-fold, reaching picomolar potency in whole-blood T cell assays and improved the peptide’s half-life in vivo [128].

Albeit with lower potency, ShK also blocks Kv3.2 channel (IC_50_~0.3 nM, [129]). Hence, the ShK therapeutic scaffold is being exploited for the development of analogs with improved selectivity profiles [130]. More selective analogs for Kv1.1 and Kv1.3 were developed by amino acid replacements with differently charged or non-natural amino acids (ShK-Dap^22^, IC_50_ 23 pM), analogs containing phospho-tyrosine (ShK-186, IC_50_ 69 pM), and phosphono-phenylalanine (ShK-192, IC_50_ 140 pM), which contain non-protein adducts and hydrolysable phosphorylated residues [115,131]. Such work suggests that selective Kv1.3 antagonists such as ShK-Dap^22^, for which structural and functional data are available, might represent promising immunosuppressant leads [106,114]. 

ShK-K-amide is an ShK analog in which an amidated lysine residue has been added to the C-terminus, resulting in potent and selective block of Kv1.3 [132]. ShK inhibits Kv1.3 and Kv1.1 channels with similar potencies (IC_50_ of 9 ± 2 pM and 23 ± 3 pM, respectively). While retaining potency (IC_50_ 26 ± 3 pM) against Kv1.3 channels, ShK-K-amide’s affinity for Kv1.1 is greatly reduced (IC_50_ 942 ± 120 pM), thus being 36-fold more selective between these two Kv1 isoforms. It is reasoned that addition of a C-terminal-amidated positive charge by the extra lysine changes the electrostatic interaction between the peptide’s C and N-termini, resulting in more favorable interaction with Kv1.3 by allowing arginine 1 to engage with the channel vestibule as well as the previously reported strongly coupled pair R29-S379 in Kv1.3-ShK [127]. However, the extra C-terminal lysine (in ShK-K-amide) disrupts binding with Kv1.1 by apparently altering K18 and R29 interactions with negatively charged residues in the channel. 

#### 3.1.2. BgK

BgK is a 37-amino-acid peptide isolated from the sea anemone *Bunodosoma granulifera*, which blocks Kv1.1, Kv1.2, and Kv1.3 channels [112]. BgK is a 37 amino acid peptide crosslinked by three disulfide bridges (C2–C37, C11–C30, and C20–C34), and free C-terminal carboxylate. Both natural and synthetic BgK inhibit binding of ^125^I-α-DTX to rat brain synaptosomal membranes with nanomolar potency [112]. Corresponding BgK residues (S23, K25, and Y26) are involved in binding to rat brain potassium channels Kv1.1, Kv1.2, Kv1.3, and Kv1.6 [133]. BgK does not select between Kv1.1, Kv1.2, and Kv1.3 channels expressed in *Xenopus* oocytes, displaying quite similar dissociating constants (K_d_ = 6 nM, 15 nM, and 10 nM, respectively [113]). BgK and ShK share 13 residues and present similar but not exact topologies (Figure 2a) ([46,126] Figure 4b). It has been shown that shortening of K25 side chain by removal of the four methylene groups dramatically decreases the affinity of BgK towards all Kv1 channels [134]. Mutations at position F6 in BgK reduce potency towards both Kv1.2 and Kv1.3 while not affecting Kv1.1, making BgK-F6A selective for Kv1.1 [135]. BgK-F6A increased miniature excitatory postsynaptic current in neurons while not affecting T-cell activation. This suggests that the Kv1.1 blockade has potential in neuro-inflammatory diseases including multiple sclerosis and stroke and BgK-F6A as a scaffold for drug design.

#### 3.1.3. BcsTx1/2

Two toxins from the venom of *Bunodosoma caissarum* were isolated and named, BcsTX1 and BcsTx2 [110]. These peptides contained the classical three disulfide bonding pattern of Kv type 1 toxins and were screened against a panel of Kv channels, displaying no affinity towards channels outside of the Kv1 subfamily. Both toxins showed differences in their Kv1 selectivity, with BcsTX1 being 10-fold selective for Kv1.2 (~30 nM) over Kv1.6 (~1.6 µM), which in turn was >10-fold selective over other Kv1 channels examined, while BcsTX1 was less selective, displaying the highest affinity for Kv1.6 but less than 10-fold greater than Kv1.1, Kv1.2, and Kv1.3 [110].

#### 3.1.4. Other Kv Type 1 Toxins

Other Kv type 1 sea anemone toxins are known to interact with Kv1 channels; however, little follow has been completed looking at their selectivity or therapeutic potential in depth. The sea anemone *Heteractis magnifica* venom contains the VDP, HmK. It is 35 amino acids long, having an identical molecular weight (MW 4055) to ShK, with 60% homology. HmK is approximately 40% identical to BgK and AsKS (Figure 4b). Partial reduction at acidic pH and rapid alkylation allowed the full assignment of the disulfide linkages (C3–C35, C12–C28, and C17–C32). HmK inhibits the binding of ^125^I-α-DTX to rat brain synaptosomal membranes with a ~1 nM K_i_ and block Kv1.2 channels and facilitates neuromuscular junction acetylcholine [114]. Alanine scanning analyses proved that six amino acids (D5, S20, and the dipeptides KY22–23 and KT30–31) are crucial for binding to rat brain Kv channels and perfectly conserved between BgK, ShK AsKS, and HmK [114]. 

AeK, isolated from *Actinia equine*, is a Kv1 channel toxin that inhibits the binding of ^125^I-α-DTX rat synaptosomal membranes in a dose-dependent manner with an IC_50_ of 22 nM [108]. The complete amino acid sequence of AeK is composed of 36 amino acid and six cysteine residues. AeK’s three disulfides are located between C2–C36, C11–C29, and C20–C33. AeK contains the canonical dyad for Kv channel block formed by K22 and Y23. AeK is similar to AsKS structurally with which it shares 86% sequence homology, 53% with BgK, and 36% with ShK (Figure 4b). However, the selectivity of this peptide has not been addressed functionally.

AsKS, or kaliseptine, is a 36-amino-acid peptide isolated from the sea anemone *Anemonia sulcata* that blocks Kv1 channels and impedes the binding of ^125^I-α-DTX to receptors in rat brain membrane [111]. AsKS shares 49% sequence homology with BgK toxin ([134] Figure 4b) but differs in two of its cysteine residues (C33 and C36) and the C-terminus. Dendrotoxin I (DTX-I) is a potent blocker of Kv1.1-, Kv1.2-, and Kv1.6-mediated currents in *Xenopus* oocytes. Despite being structurally dissimilar, AsKS appears to share a receptor site in Kv1 channels with the kalicludines (AsKC) and DTX-I. The simple comparison on the capacity of AsKS inhibition for Kv1.2 channel with inhibition ^125^I-α-dendrotoxin binding to neuronal membranes should be followed with a more in-depth investigation.

The mature AETxK peptide from *Anemonia erythraea* is 34 residues long; six cysteines are paired to form three disulfide bridges (C2–C34, C11–C27, and C16–C31) and presents a canonical Kv channel-blocking dyad comprised of K21 and Y22. AETxK is 59% and 65% homologous to ShK and HmK respectively, whereas it shares 41–44% sequence homology to all other type 1 anemone toxins (Figure 4b). AETxK blocks ^125^I-α-DTX binding to rat synaptosomal membranes with an estimated IC_50_ of 91 nM [109]. No electrophysiological or related functional data has been reported for this peptide; therefore, its selectivity is unknown [109].

These VDP are all similar to the well-studied ShK; thus, in-depth selectivity profiling is required for these toxins on both homomeric and heteromeric Kv1 channels. These toxins have the potential as scaffolds for therapeutics targeting of autoimmune disorders, stroke, diabetes, multiple sclerosis, and others. None of these toxins have been tested on hetero-tetrameric Kv1 channels and, with their divergent affinities across the Kv1s, may provide interesting routes to discovering selective compounds for various heteromeric Kv1 channels.

### 3.2. Kv Type 2 Anemone Toxins

Kv type 2 anemone toxins all contain a kunitz-type motif and function as both protease inhibitors and Kv channel blockers. They were first isolated from sea anemones *Anemonia sulcata* and named kalicludines (AsKC1–AsKC3) [130]. However, other Kv type 2 inhibitors have been discovered including APEKTx1 (*Anthopleura elegantissima*) [136] and SHTXIII (*Stichodactyla haddoni*) [117]. When compared to Kv type 1, Kv type 2 anemone toxins typically have lower affinity towards Kv1 channels, making their biological role unclear. They may act to paralyze prey through their dual action protease/Kv channel activity, to provide protection for prey/predator proteases (or, in a similar fashion, to provide protection for their own venom components when injected into their prey/predator), or to act to regulate digestive mechanisms [137]. AsKC1-3 (kalicludines 1–3), are composed of 57–60 amino acid residues with protease inhibitor activity [111]. A mutation at position 19 lowers their inhibitory effect that linked to the sequence homology of BPTI, protein Kunitz-type protease inhibitors [104]. AsKC1 and AsKC2 contain the fully conserved dyad K5/L9 responsible for competing with the DTX-I site in Kv1 channels. They share ~40% amino acid homology with other toxins from venomous animals, such as DTX and BPTI, but AsKC1 and AsKC2 have different specificity from AsKC3 [111,130]. However, further studies should be conducted on the selectivity of these toxins. 

In contrast, APEKTx1 has an in-depth study into its selectivity across a variety of ion channels. This study revealed potent activity against Kv1.1 (0.9 nM) with >1000-fold selectivity over other Kv1 channel members, making it an excellent probe into Kv1 channelopathies.

### 3.3. Kv Type 3 Anemone Toxins

Kv type 3 anemone toxins contain a β-defensin-fold characterized by a short helix or turn followed by a small twisted antiparallel β-sheet. β-defensin are antimicrobial peptides, and anemones have weaponized them as neurotoxins [138] to target not just Kv channels but also Nav and acid sensing ion channels (ASIC) [139,140]. Five Kv type three toxins are shown to have affinity towards Kv1s although very modestly. These include BDS-I (blood depressing substance I from *Anemonia sulcata*), APETx1/2/4 (*Anthopleura elegantissima*), and PhcrTx2 (*Phymanthus crucifer*). BDS-I was first characterized as an antihypertensive and antiviral compound [141]. It was later shown that BDS-I inhibited Kv3.1, Kv3.2, and Kv3.4 with nanomolar concentrations, with only a week inhibitory effect of Kv1.1–5 [116]. BDS-1 has also been shown to inhibit voltage-gated sodium channels in the nanomolar range [142]. Similar to BDS-I, APETx1, 2, and 4 have similar issues with selectivity towards Kv1s. APETx1 and APETx4 are more selective for hERG [118,143], while APETx2 is selective for ASIC channels [138]. PhcrTx2 showed little inhibition of the total DRG Kv channels currents (IC_50_ 6.4 µM) or heterologously expressed Kv1 channels [119]. Currently, no Kv type 3 anemone toxin is specific for Kv1s, suggesting this family of toxins may not be Kv1 therapeutics of potential pharmacological agents. However, future studies will be interesting to address if the Kv type 5 anemone toxins are all antimicrobial and, if so, do they target prokaryotic potassium channels.

### 3.4. Kv Type 4 Anemone Toxins

Kv type 4 anemone toxins are characterized by a novel fold called boundless β-hairpin. SHTX I (28 residues) and SHTX II (an analog of SHTX I, 28 residues), based on their structure, has been shown that SHTX I and II have only been shown to inhibit binding of ^125^I-DTX [117] with no channel blocking specificity described. Further studies are required to ascertain selectivity and potency towards Kv1s to assess their future potential as therapeutics or pharmacological tools

### 3.5. Kv Type 5 Anemone Toxins

Kv type 5 anemone toxins contain the ubiquitous, inhibitor cysteine knot (ICK) motif. BcsTx3 is a 50-amino-acid peptide and Kv toxin extracted from the venom of the sea anemone *Bunodosoma caissarum* with a molecular weight of 5710.52 Da and four disulfide bridges. High identity (65.3% and 63.3%) with BscTx3 and identical positions of cysteine residues have been found in toxins isolated from *Nematostella vectensis* and *Metridium senile*. Another Kv type 5 anemone toxin, PhcrTx1 (*Phymanthus crucifer*), has low affinity modulatory actions on Kv1 channels; however, its main target has been shown to be ASIC channels [120].

Activity investigation of BcsTx3 has been done by Reference [110] on 12 cloned voltage-gated potassium channels and 3 voltage-gated sodium channels. It was shown that BcsTx3 blocks Kv channels Kv1.1, Kv1.2, Kv1.3, Kv1.6, and Shaker IR (inactivation removed) and did not show any activity on sodium channels. The blockage activity of BcsTx3 is not voltage dependent, and the binding site is located at the extracellular side. The lysine and tyrosine functional dyads are absent although present in most of the pore blocker toxins. BcsTx3 binds to Shaker IR through multipoint interaction due to existence of two putative dyads (R5-Y6 and R39-Y40). The evolution of the neurotoxin gene family has been followed by sequencing of the entire genome of *N. vectensis*. It has been shown that peptides in sea anemones responsible for blocking Kv channels evolved at least five times independently but that adaptive evolution took place in a common ancestor [144]. It remains to be seen if any Kv type 5 anemone toxins will provide selectivity within the Kv1 channel family, and thus, any potential as a therapeutic scaffold so far is limited. However, the ICK motif is highly stable, resistant to denaturation and proteolysis [145], which make excellent potential therapeutics if selectivity can be conferred.

### 3.6. Kv Type 6 Anemone Toxins

Kv type 6 anemone toxins are the shortest of Kv-type sea anemone toxins and contain a proline-hinged asymmetric β-hairpin fold. AbeTx1 is a toxin with a unique primary structure isolated from nematocysts of the sea anemone *Actinia bermudensis*. It is short flexible random-coil-like conformational peptide chain of 17 amino acids with a tendency to form β-sheet (aromatic or aliphatic amino acids are not present, but it contains a high proportion of Lys and Arg, and two disulfide bridges between C1–C4 and C2–C3) [121].

The activity of AbeTx1 was tested on 12 subtypes of Kv channels (Kv1.1–Kv1.6; Kv2.1; Kv3.1; Kv4.2; Kv4.3; Kv11.1; and Shaker IR) and three voltage-gated sodium channels (Na_V_1.2, Na_V_1.4, and BgNa_V_). It has been shown that AbeTx1 is selective for Shaker-related K^+^ channels and is capable of inhibiting K^+^ currents by blocking the K^+^ current of Kv1.2 or by altering activation of Kv1.1 and Kv1.6 channels. AbeTx1 showed no activity on sodium channels, but the same concentration (3 μM) inhibits the current of Kv1.1, Kv1.2, Kv1.3, Kv1.6, and Shaker IR channels [121]. It is known that the mechanism of Kv channel toxins is multipoint interaction binding to ring based amino acid, and due to the presence of six such amino acid rings (R1, R9, R11, K3, K7, and K13), AbeTx1 toxins interact with Kv1.1 and Kv1.6 channels. Electrophysiological experiments were performed to determine the mechanism of action, and it has been discovered that probably AbeTx1 toxin binds on the outer side of Kv1.2 and Kv1.6 channels since its effect is reversible (current was recovered). Competitive binding experiments with TEA showed that binding sites for both ligands are not overlapping completely on Kv1.1, which is not the case with Kv1.6 channels due to less voltage dependence of blockage of Kv1.1 with membrane depolarization [121].

Also, alanine point-mutated analogs were tested on Kv1.1 and Kv1.6 channels. Six synthetic analogs were used to test a multipoint interaction of the toxin to the channel’s binding site, and it has been shown that loss of the side chains will lead to decreasing activity of analogs [121]. 

## 4. Non-Peptidyl Kv1 Channel Inhibitors

### 4.1. Gambierol

One of the non-peptidyl toxins from the ciguatera group (CigTXs) is gambierol (Figure 5), marine polycyclic ether toxin isolated from marine dinoflagellate *Gambierdiscus toxicus* showing acute toxicity in mice (LD_50_ = 50 mg/Kg, ip; [146]). These CigTXs are accumulated throughout the marine food chain, causing hypotension, bradycardia, respiratory difficulties, and paralysis [147]. Although CigTXs are toxic for Nav channels, gambierol inhibits only Kv1 potassium channels [147,148].

Gambierol is lipophilic and can pass through the cell membrane. This was confirmed as Gambierol inhibited closed channels no matter which side the toxin was applied [148]. Gambierol has heptacyclic and tetracyclic analogs that are specific inhibitors for the Kv1.2 channel expressed in CHO cells with IC_50_s of 0.75, 7.6, and 28 nM, respectively [146]. However, the dose-dependent leak current induced by these compounds might be behind their cytotoxic effect. Screening of Gambierol against other Kv channels revealed that Kv2 and Kv4 were insensitive to 1 μM gambierol but fully repressed Kv3.1 channels [149]. The same study showed ~70% inhibition of K^+^-mediated currents by Kv1.4 using 100 nM Gambierol.

According to Kopljar, et al., 2009 [149], Kv3 channels inhibition occurs when channels are deactivated, suggesting a mechanism related to gating modification. Swapping of the S5–S6 linker between Kv3.1 and Kv2.1 channels gave no differences in selectivity of gambierol to either of the channels, indicating that gambierol is not an external pore blocker. Kv1 and Kv3 channels contain threonine residues in the inner permeation pathway, while the insensitive Kv2 and Kv4 channels have valine at the same position. After the replacement of threonine with different moieties, it was confirmed that hydrogen bonding capable amino acids (serine and lysine) contribute to the high affinity of gambierol to Kv3.1 channels. The T427 residue between the S5 and S6 segments of Kv3.1 channels interacts with one of the ether oxygens of the toxin, thereby inhibiting permeation of K^+^ ions and stabilizing the closed state. 

### 4.2. Aplysiatoxin Derivatives

Marine cyanobacteria are a source of many toxins, including a recently discovered group of Kv1.5 blockers called Aplysiatoxins (ATXs). ATXs and related analogs, namely Oscillatoxins and nhatrangins (Figure 6), are 27 bioactive dermatoxins polyketide compounds that were isolated from several marine cyanobacteria species with antiproliferative activity, tumor-promoting properties, proinflammatory actions, and antiviral activity (see Reference [150]). According to the structural characteristics, ATXs were divided into three categories: (1) the ABC tricyclic ring systems with carbon numbers of 6/12/6, 6/10/6, and 6/6/6 (e.g., Debromoaplysiatoxins and neo-debromoaplysiatoxins) [151]; (2) the AB spirobicyclic ring system (e.g., Oscillatoxin D) [152], and (3) acyclic structures such as Nhatrangins [153]. 

In addition to their various structures, ATX derivatives exhibit selectivity and potency against Kv1.5 channels, a possible pivotal target for new treatment of atrial tachyarrhymias with minimal potential for deleterious side effect [151,154,155]. Although further studies are still ongoing to establish the Kv1.5 inhibition mechanism by these various bioactive compounds, researchers suggested two mechanisms. One is the direct ion channel modulation by direct blocking of the pore. The other proposed mechanism is the indirect modulation of the Kv channel by activating protein kinase C [154].

## 5. Kv1-Active Toxins in Research and Drug Discovery

Venom-derived toxins have been paramount in the identification and study of ion channels. Neuronal Kv1 heteromeric complexes were first recognized thanks to snake dendrotoxins and were identified by isoform-specific antibody fractionation [36,37,40]. We have learned that the majority of Kv1 channel complexes present in the nervous system are hetero-tetrameric combinations of Kv1 α-subunits, while only a small fraction of channels are homo-tetramers (e.g., neuronal: Kv1.2 and 1.4; immune system Kv1.3). The composition, stoichiometries, and subunit arrangements of Kv1 complexes expressed in different tissues and cells remains to be fully identified. As seen in the sections above, many of the venom-derived toxins published have not been screened against homomeric Kv1 channels and thus lack information of biological potential. To understand the therapeutic potential of these toxins, it is necessary to study their effects on relevant heteromeric Kv channel complexes. This requires efforts that include but are not limited to the generation and functional characterization of concatenated (or tandem) hetero-tetramers [156,157], as proxies of the physiological targets, in bioactive-driven molecular tool development and drug discovery. 

By examining venom-derived peptides on hetero-tetrameric Kv1, it has recently been discovered that some VDP display significantly higher potencies for heteromeric Kv1s than those with the same contributing homomeric channel isoforms. For example, testing of conotoxin κM-RIIIJ over 12 different heterodimers containing Kv1.2 subunits, Cordeiro et al. showed that RIIIJ was most potent against Kv1.1/Kv1.2 heterodimers without apparent regard for their arrangement, showing significant discrimination against other heterodimeric constructs including those formed by association of Kv1.2 with Kv1.5 or Kv1.6. Further functional analyses showed that RIIIJ was indeed ~100 more potent against hetero-tetramers made of three copies of Kv1.2 α-subunits and one of either Kv1.1 or Kv1.6 in a 3:1 stoichiometry [82]. This detailed functional and biochemical characterization then enabled the use of RIIIJ as a molecular tool for the classification of live large DRG neurons into six discrete functional populations [83]. An approach that can be exploited to accelerate the classification and study of other cell types within mixed population.

The thorough functional characterization of Conkunitzin-S1 allowed the identification of Kv1.7 channels in pancreatic beta cells as contributors in glucose-stimulated insulin secretion [12]. As shown for the RIIIJ/Kv1.2 interactions, the presence of the Kv1.7 α-subunit confers sensitivity to Conk-S1 to a Kv1 hetero-tetramer. In contrast to RIIIJ, Conk-S1 (and Conk-S2) appears to be able to discriminate different Kv1 targets based on their relative positioning [12]. Thus, hinting to the Conks’ potential as molecular tools in the study of heteromeric Kv1 channels in native tissues. 

The thorough characterization and in vivo study of Conk-S1 was possible due to the production of high yields by recombinant expression of the peptide in *E. coli* [52], which guarantees inexpensive production. Much like monomeric insulin, Conk-S1 is a small peptide (<10 kDa) in which therapeutic potential for the treatment of hyperglycemic disorders is supported principally by its effectiveness, in vivo and ex vivo, to enhance insulin secretion and lower glucose levels in a strictly glucose dependent manner because the targeted Kv1.7 channel opens at depolarized potentials that are achieved upon increases in blood glucose (i.e., postprandial). Consequently, Conk-S1 treatment appears to modulate pancreatic β-beta cell excitability only at stimulatory glucose concentrations where bursting electrical activity is observed but not at low/basal glucose concentrations, eliminating the risk of hypoglycemia [12]. This is advantageous because the current drugs used to ameliorate diabetes are effective at lowering blood glucose but do so regardless of the basal conditions. Hence, acute, delayed, and persistent hypoglycemia constitutes the most frequent adverse effect associated with sulfonylureas (K_ATP_ channel inhibitors) and insulin-based therapies, obliging frequent monitoring of blood glucose concentrations [158]. Furthermore, intraperitoneal Conk-S1 injections neither affected basal glucose levels nor produced adverse cardiovascular or neurological side effects in vivo [12], highlighting the safety of targeting Kv1.7 channels. Nevertheless, the identification and validation of biological target(s) as well as prediction of biological activities of Conk-S1 must be verified in alternative systems including animal models in order to ascertain the mechanism of action behind Conk-S1’s pro-insulinogenic effects. Further, Structure-Activity Relationship (SAR)-aided computational work would be useful to aid functional refinements in the selectivity and potency of Conk-S1 for its development as a therapeutic to minimize potential side effects and to decrease production costs.

Much has been done in the case of the anemone toxins. For instance, ShK is considered the most potent blocker for Kv1.3 channels with an IC_50_ of 10 pM [106]. However, it also potently blocks Kv1.1, 1.4, and 1.6 channels [131,159]. Kv1.3 has been shown to be a potential target of immune-modulators; hence, in order to enhance its selectivity over other targeted Kv1 isoforms, analogs of ShK have been generated [115]. These efforts paved the way for the development of a leading blocker called ShK-186, which has a 100-fold improvement in selectivity for Kv1.3 over Kv1.1, 1.4, and 1.6 channels [160]. Currently known as dalazatide, it successfully passed phase 1 clinical trials in 2016 and entered phase 2 in 2018 for the treatment of several autoimmune diseases like inclusion body myositis, lupus, multiple sclerosis, psoriasis, rheumatoid arthritis, type 1 diabetes, and inflammatory bowel diseases. Promising newer generation, ShK analogues are currently under development [102].

## 6. Challenges and Outlook

In this review, a conscious attempt was made to provide an overview of those Kv1-targeted marine bioactives for which functional data including potency and selectivity has been reported. The availability of such information has allowed their development as molecular tools, as are the cases of κ-RIIIJ and Conk-S1, and therapeutically promising pharmacological scaffolds like ShK or Conk-S1. The wealth of marine bioactive molecules targeting Kv1 channels is immense, yet the vast majority has been characterized only at the sequence level. The collection of marine toxins presented here was focused around Kv1 channel modulatory activity. However, the selectivity profiles of most marine toxins, both the peptides and non-peptides, are absent or inadequate; hence, further work is necessary to assess their real potential as research tools and therapeutics. 

Competition binding data is customarily reported for venom-derived compounds, but this is at best indicative of their potential bioactivity. Comprehensive functional profiling is fundamental to attest to the potential of the abundance of toxins identified, albeit it is not achieved without substantial challenges. First, the primary sequence of venom derived peptides is used to predict their 3D structure and to guide inference of potential targets. Unfortunately, similar scaffolds are often used to target across families of ion channels and enzymes; therefore, functional verification is an absolute requirement. Second, most selectivity screens to determine specificity of Kv1-targeted compounds are limited to their functional assessment in homo-tetrameric Kv1 channels assembled upon expression of a single α-subunit in heterologous systems. Most native Kv channels are not homomeric but heteromeric complexes formed by up to four different α-subunits, as is the case of the CNS and PNS Kv channels, and thus, natural bioactive species such as marine toxins and VDPs have evolved to target heteromeric combinations. These natural molecules have been selected through evolution to serve as molecular tools for the study of native Kv channels. Furthermore, the study of the molecular determinants guiding marine bioactive targeting of heteromeric Kv channels would aid the design of pharmacological agents for the treatment of various channelopathies.

κ-conotoxins, such as RIIIK, RIIIJ, Conk-S1, and Conk-S2 toxins, have been shown to discriminate among targets based on heteromeric composition and order of connectivity [12]. Naturally evolved K^+^ channel-targeted ligands may become instrumental in the study of heteromeric Kv1 channels in live biological systems allowing facile determination of composition and physiological function. Efforts must be made towards bioactivity determination in hetero-multimeric Kv1 channels in heterologous systems and by coupling functional studies with single cell transcriptomics/proteomics of primary cultures. The findings from such work would provide invaluable to the development of leading drugs against heteromeric channels associated with Kv1 channelopathies [25].

VDP inhibitors select among heteromeric Kv1 channel targets according to their α-subunit identities, their stoichiometry, and their arrangement by binding across monomeric boundaries. This could account for the diversity of selectivity “fingerprints” observed in native cells/tissues and highlights marine toxin relevance as molecular tools and pharmacological applications. The potency and selectivity of bioactives found in nature provides a wealth of scaffolds with therapeutic potential. From one side, the molecular understanding of the ion channel pore structure was revealed by using natural peptide-based toxins as molecular probes. On the other hand, better understanding of the Kv1 channels as drug targets for the treatment of disease is crucial for developing promising therapies. 

Comparatively speaking, few marine toxins have been functionally explored in enough depth to be considered for research or clinical purposes. Only a few labs in the world have full capabilities to perform all: discovery, synthesis/production, functional characterization, and experimentation in animal models. Therefore, collaboration between expert labs from each and all disciplines involved is an absolute requirement for the advance of the field. 

Despite the high interest in the discovery of novel marine toxins, many challenges exist in advancing them into therapeutically active compounds. Amongst those challenges, the development of adequate transferable human assays to assess compound’s selectivity and off-target activity is paramount. Selectivity and SAR screens are time and labor intensive, requiring large amounts of pure material. For example, synthesis and purification of biologically active peptides are ridden with numerous potential pitfalls such as the selection of the optimal prokaryotic or eukaryotic expression host (bacteria, yeast, or mammalian cell lines). The production of recombinant proteins has predominantly used bacterial expression due the ability to generate high protein yields from large volume cultures of fast-growing, low-cost bacteria [161,162]. However, when the protein products involve eukaryotic posttranslational modifications, mammalian or insect cells become the system of choice. Often, functionally, many marine peptides require proper folding supported by their cysteine connectivity, which can curtail yields of active proteins [163] and/or incorporate posttranslational modifications critical for their activity. Despite the advantages of eukaryotic expression systems, higher production costs preclude their use [164]. Recently, periplasmic peptide expression in *E. coli* has made use of the oxidizing environment to produce folded peptides (i.e., Reference [165]). While this has the potential to speed up protein production, it is limited as targeting across the cytoplasmic membrane can substantially limit periplasmic yields [166,167]. Marine non-peptidyl compounds also pose challenges inherent to their chemistry and organisms that produce them [168]. However, once production of marine compounds is achieved, their stability, formulation, delivery, and antigenicity need to be overcome. Despite these limitations, the potential therapeutic abilities of marine natural products are unquestionable. Successful examples include Zinconotide (Prialt, sever chronic pain), Cytarabine (Cytosar-U, chemotherapy medication), Vidarabine (Vira-A, antiviral drug), Brentuximab Vedotin (Adcetris, chemotherapy medication), Eribulin Mesylate (Halaven, chemotherapy medication), Omega-3-acid ethyl esters (Lovaza, diet and exercise drug which reduces triglycerides), Trabectedin (Yondelis, chemotherapy medication), Fludarabine Phosphate (Fludara, chemotherapy medication), Nelarabine (Arranon, chemotherapy medication), and Iota-carrageenan (carragelose, Antiviral drug), with ~30 in various clinical phase I-IV trials (for a recent review, see Reference [169]).

Marine toxins remain a relatively under-explored source of bioactives targeting Kv1 channels. Technological advances in transcriptomics and proteogenomic will enable the expedited identification of novel marine toxin repertoires, will explore the diversity of their function based on known peptide scaffold, and will understand the relationships of structure-function aspects of these toxins with much more still that remains to be discovered. Importantly, bioactive function can only be determined experimentally. The use of marine toxins and novel, integrative strategies provide powerful approaches to functionally define specific cellular types underlying physiology in health and diseased states. Given the laborious nature of electrophysiological recordings, high-throughput functional assessment platforms involving automated patch clamp are fundamental for substantial output increase in the discovery of novel ion-channel targeting molecules and the advancement of the venom-derived drug discovery field.

## Figures and Tables

**Figure 1 marinedrugs-18-00173-f001:**
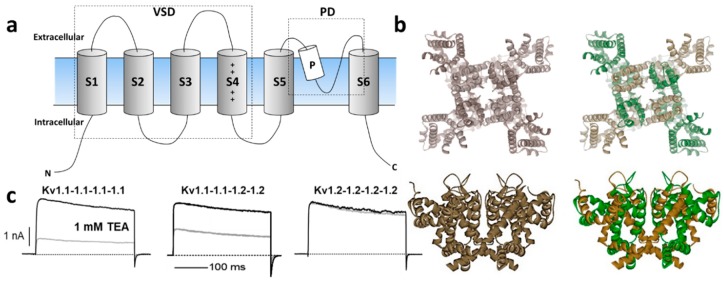
(**a**) Schematic illustration of K_V_ channel membrane topology depicting the 6 transmembrane subunits including the voltage sensing domain (voltage-sensing domain (VSD): S1–S4) and the pore domain (PD) between S5 and S6 segments. (**b**) Top and side views of representative homomeric and heteromeric Kv1 channels based on the crystal structure of Kv1.2 channels [(Protein Data Bank number, PDB: 2A79)] [13]. (**c**) Current trances of homomeric Kv1.1 (left) and 1.2 (right) channels and their heteromeric combination (middle) revealing distinct sensitivity to the classical pharmacological tool tetraethylammonium (TEA) [23,24].

**Figure 3 marinedrugs-18-00173-f003:**
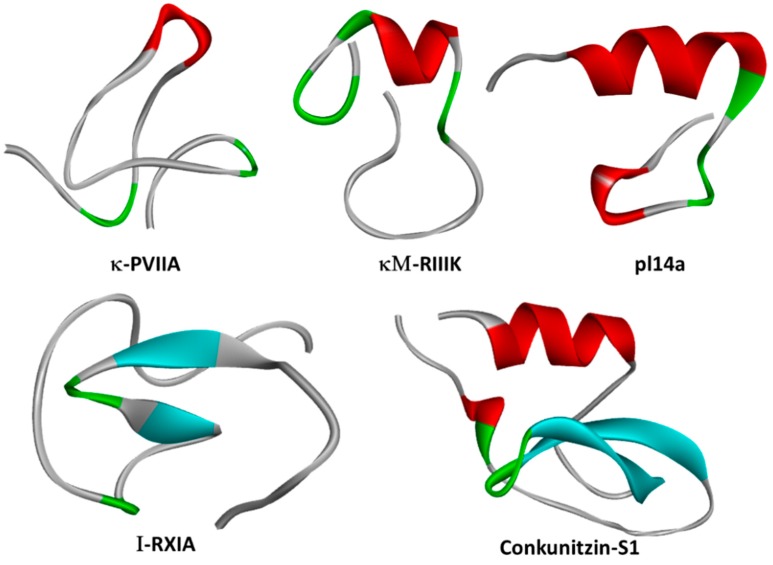
Structures of representative cone snail venom-derived peptide toxins κ-PVIIA (PDB: 1AV3, [75]), κM-RIIIK [51], pl14a (PDB: 2FQC, [76]), I-RXIA (PDB: 2JTU, http://www.rcsb.org/structure/2JTU), and Conkunitzin-S1 (PDB: 2CA7, [52]): β-sheets are in cyan, and α-helices are in red.2.1. κM-RIIIK.

**Figure 4 marinedrugs-18-00173-f004:**
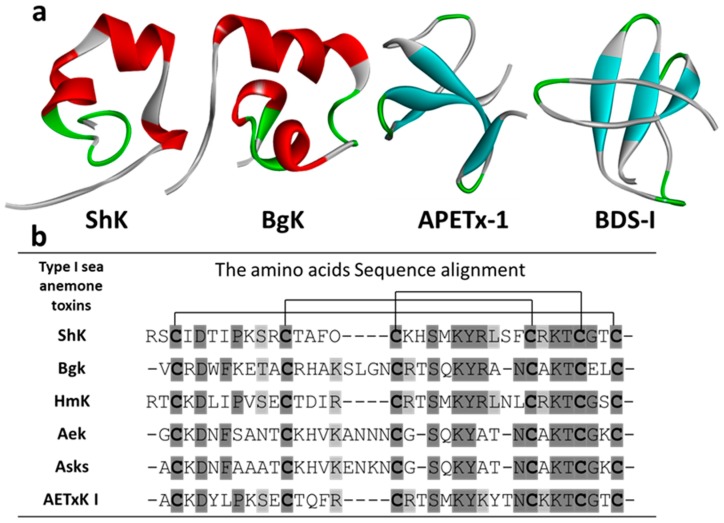
(**a**) Structures of sea anemone peptide toxins ShK (PDB: 1ROO, [53]), BgK (PDB: 1BGK, [46]), APETx-1 (PDB: 1WQK, [122]), and BDS-I (PDB: 2BDS, [123]): The location of the disulfide linkages are shown in green, beta-sheets are in blue, and alpha-helices are in red. (**b**) Sequence alignment of type 1 sea anemone K_V_-toxins according to their cysteine framework with the pairings indicated by the lines linking them: Amino acid identity (dark shade) and similarities (light shade) are shown [110].

**Figure 5 marinedrugs-18-00173-f005:**
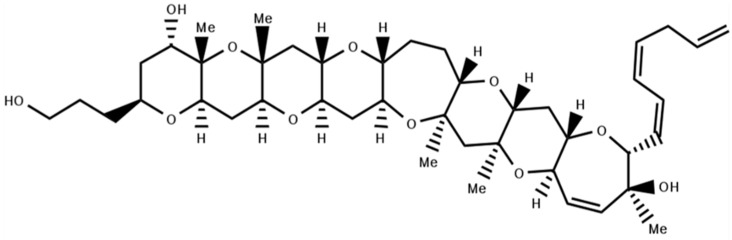
Structure of Gambierol toxin showing the eight polyether rings [129]: Me indicates a methyl group.

**Figure 6 marinedrugs-18-00173-f006:**
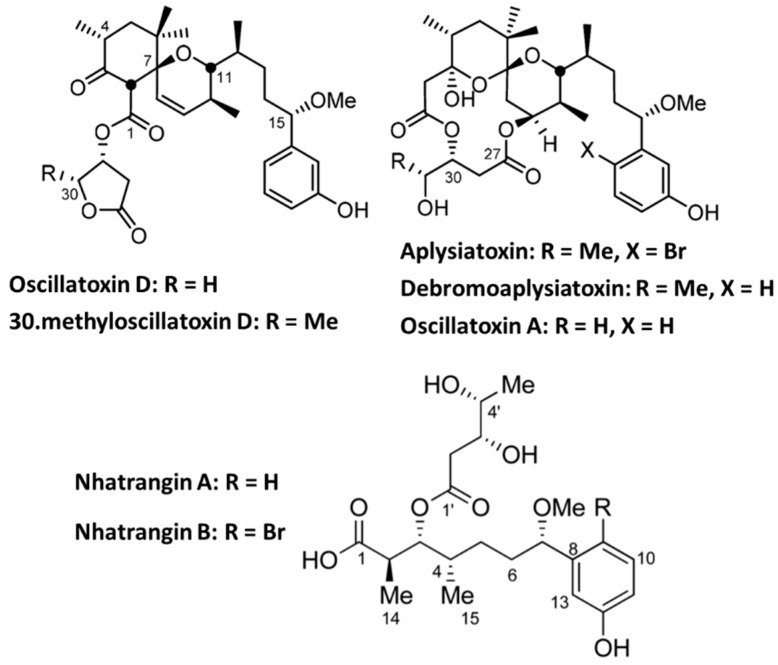
Structure of representative Aplysiatoxin derivatives from References [152,153]: Me indicates a methyl group.

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
