# Peer review of "Marine Toxins Targeting Kv1 Channels: Pharmacological Tools and Therapeutic Scaffolds"

_marinedrugs, 2020, doi:10.3390/md18030173_

Round 1
Reviewer 1 Report
See attached comments.

Author Response
Many thanks for the positive comments and input that has helped us improve the quality of our manuscript and hopefully extend its reach and impact.
-All edits, corrections and additions have been tracked.
Point 1: Delete the second paragraph: “The introduction should briefly place the study in a broad context and highlight why it is important. It should define the purpose of the work and its significance. The current state of the ………See end of the document for further details on references.”
Response: Apologies for the oversight, the lingering paragraph from the template form has been removed.
Point 2: Please include a figure defining the sites of binding on Kv1 channels of different marine toxins. Docking models would be useful for toxins where information is known.
Response: We appreciate the suggestion and have incorporated the suggested figure as Figure 2 including the major bidding sites for marine toxins and the gating modifier Hanatoxin in the Kv1.2-2.1 chimera crystal structure. Docking models for selected blockers with their highest affinity targets were included to represent the major modes of blockade against Kv1 channels.
Point 3: “The identified genes for K channel a -subunits are classified into four subfamilies: Kv1 (Shaker), Kv2 (Shab), Kv3 (Shaw), and Kv4 (Shal) families.” This statement is wrong. As stated at the beginning of paragraph 4 (line 81) KV channels are classified into 12 sub-families (and 46 Kv genes) (https://www.guidetopharmacology.org/GRAC/ReceptorFamiliesForward?type=IC). Please correct.
Response: The statement has been revised within lines 81-84.
Point 4: When describing hetero-multimeric and/or concatenated Kv1 subunits (shown in Figure 1C), besides the references included, please also cite: J Biol Chem. 1992;267:23742-5; Neuron 1992;8:493-7; Receptors Channels. 1995;3:263-72; Biochem J. 2013;454:101-8.
Response: Many thanks, the missing references were inserted were relevant. These references are [14, 15, 148 and 149].
Point 5: When describing the structural basis for C-type inactivation, besides the references included, please also cite: Elife. 2018;7. pii: e37558; J. Gen. Physiol. 2013;141:151–160; Nature Structural & Molecular Biology 2017; 24:857–865.
Response: Many thanks, the missing references were inserted were relevant. These references are [19-21].
Point 6: When describing ShK, besides the references included, please also cite: Toxicon 1995;33:603-13; Nat Struct Biol. 1996;3:317-20. Please also include ShK-EWSS (FEBS J. 2015;282:2247-59) and Amgen’s ShK-PEG and ShK-peptibodies (J Med Chem. 2015;58:6784-802).
Response: The suggested references have been included in the text where ShK was described. These references are [116, 118, 119, and 120].
Point 7: Structures of type 1 sea anemone toxins (ShK and BgK) are shown. Please also show structures of other types of KV1-modulating sea anemone toxins (e.g. PDB: 1WQK; 2BDS).
Response: Figure 3a was updated with two additional sea anemone toxins, and the figure legend was updated accordingly.
Point 8: In the discussion of the therapeutic potential of marine toxins, please highlight the challenges faced in the conversion of marine toxins into therapeutics: selectivity, off-target activity, stability, formulation and delivery (ShK-186 has a surprisingly long circulating half-life after subcutaneous injection), antigenicity, manufacture of cysteine-rich peptides, use of unnatural amino acids, peptide-drug conjugates. Since the first author described the therapeutic potential of Conkunitzin-S1 for diabetes mellitus, ConkunitzinS1 could be used as the exemplar to explain the steps that would be needed to convert it into a drug. Please also highlight the potential advantage of Conkunitzin-S1, a glucose-dependent insulin secretagogue via blockade of Kv1.7, over sulfonylureas, glucose-independent secretagogues that target SUR/KIR6.x. The authors may also wish to speculate on which other Kv1-modulating marine toxins could be developed into therapeutics.
Response: We thank you for the constructive suggestions. We have implemented two extended paragraphs discussing the points raised. Please see the added paragraphs in the "Kv1-active toxins in research and drug discovery" lines (658-660 and 684-706) and "Challenges and Outlook" (Lines 720-723 and 764-790) sections. Accordingly, these paragraphs were supported with References 150-161, which are now included in the Reference list.
Reviewer 2 Report
This is a well-written review regarding the modulation of Kv1.1 channel by toxins. The authors presented lots of information regarding this topic and will give readers a big picture for toxin regulation of Kv1.1 channels as well as other ion channels. I only have a minor concern.
In lines 42 to 49, this paragraph should not be in the text and need to be deleted.
Author Response
Many thanks for the positive comments and input from the reviewers that has helped us improve the quality of our manuscript and hopefully extend its reach and impact.
-All edits, corrections and additions have been tracked.
-In lines 42 to 49, this paragraph should not be in the text and need to be deleted.
Response: Apologies for the oversight, the lingering paragraph from the template form has been removed.